# A Comparative Review of Hot and Warm Mix Asphalt Technologies from Environmental and Economic Perspectives: Towards a Sustainable Asphalt Pavement

**DOI:** 10.3390/ijerph192214863

**Published:** 2022-11-11

**Authors:** Abdalrhman Milad, Ali Mohammed Babalghaith, Abdulnaser M. Al-Sabaeei, Anmar Dulaimi, Abdualmtalab Ali, Sajjala Sreedhar Reddy, Munder Bilema, Nur Izzi Md Yusoff

**Affiliations:** 1Department of Civil and Environmental Engineering, College of Engineering, University of Nizwa, P.O. Box 33, Nizwa PC 616, Ad-Dakhliyah, Oman; 2Centre for Transportation Research, Department of Civil Engineering, Faculty of Engineering, University of Malaya, Kuala Lumpur 50603, Malaysia; 3Department of Civil and Environmental Engineering, Universiti Teknologi PETRONAS, Bandar Seri Iskandar 32610, Perak, Malaysia; 4College of Engineering, University of Warith Al-Anbiyaa, Karbala 56001, Iraq; 5School of Civil Engineering and Built Environment, Liverpool John Moores University, Liverpool L3 5UX, UK; 6Department of Civil Engineering, Faculty of Engineering and Applied Science, Memorial University of Newfoundland, St. John’s, NL A1B 3X5, Canada; 7Department of Civil Technology, College of Science Technology-Qaminis, Qaminis, Libya; 8Department of Civil Engineering, Universiti Kebangsaan Malaysia, UKM, Bangi 43600, Malaysia

**Keywords:** hot-mix asphalt, warm-mix asphalt, life cycle assessment, gas emission, energy consumption, global warming, sustainable pavements

## Abstract

The environmental concerns of global warming and energy consumption are among the most severe issues and challenges facing human beings worldwide. Due to the relatively higher predicted temperatures (150–180 °C), the latest research on pavement energy consumption and carbon dioxide (CO_2_) emission assessment mentioned contributing to higher environmental burdens such as air pollution and global warming. However, warm-mix asphalt (WMA) was introduced by pavement researchers and the road construction industry instead of hot-mix asphalt (HMA) to reduce these environmental problems. This study aims to provide a comparative overview of WMA and HMA from environmental and economic perspectives in order to highlight the challenges, motivations, and research gaps in using WMA technology compared to HMA. It was discovered that the lower production temperature of WMA could significantly reduce the emissions of gases and fumes and thus reduce global warming. The lower production temperature also provides a healthy work environment and reduces exposure to fumes. Replacing HMA with WMA can reduce production costs because of the 20–75% lower energy consumption in WMA production. It was also released that the reduction in energy consumption is dependent on the fuel type, energy source, material heat capacity, moisture content, and production temperature. Other benefits of using WMA are enhanced asphalt mixture workability and compaction because the additives in WMA reduce asphalt binder viscosity. It also allows for the incorporation of more waste materials, such as reclaimed asphalt pavement (RAP). However, future studies are recommended on the possibility of using renewable, environmentally friendly, and cost-effective materials such as biomaterials as an alternative to conventional WMA-additives for more sustainable and green asphalt pavements.

## 1. Introduction

The environmental concerns of global warming are among the most severe issues facing human beings. A contributing factor in the flexible pavement, primarily using HMA, is significant fuel and energy consumption, resulting in pollutant emissions [1]. In contrast, WMA technology was developed to meet sustainability’s economic and environmental needs. However, humans have long been constructing flexible pavements to ensure smooth and durable road pavements [2]. Hence, the pace of road construction has been increasing globally, and 12 million km of roads were constructed in 2000; it is projected that 25 million km of roads will be built by 2050 globally [3]. For example, in Malaysia, the Public Work Department (JKR) reported that there was 237,022 km of roads in 2017 [4]. The construction of about 90% of the world’s paved roads uses asphalt mixture, and the remaining 10% are other types of pavement [5]. The increasing traffic volume necessitates using asphalt binders and mixes with enhanced properties to ensure the durability of asphalt pavements [1]. Generally, asphalt mixtures comprise three main materials, namely aggregate, asphalt binder, and filler. The aggregates and filler make up approximately 94–96% of the total mixture weight, and the remaining 4–6% is asphalt binder. These materials are heated to high temperatures of 150 and 180 °C to ensure proper aggregate coating by the asphalt binder and adequate workability of the mixture. This process consumes a large amount of energy and emits gases. One of the biggest problems faced by the world is global warming [6]. The high pace of transportation contributes to the emissions of large amounts of greenhouse gases that cause global warming [2,7,8,9]. According to the Inventory of U.S. Greenhouse Gas Emissions and Sinks, transportation contributes about 27% of total U.S. GHG emissions in 2020 [10]. The Kyoto Protocol adopted in 1977 aims to develop technologies that reduce the emissions of gases that cause global warming. Therefore, the road construction industry has adopted various techniques to control and reduce the emission of greenhouse gases, and one of them is the warm-mix asphalt (WMA). Generally, there are four types of asphalt mixtures, depending on their production temperature. (i) The cold mix asphalt (CMA) produced at 0–30 °C; (ii) the half-warm mix asphalt (HWMA) produced at 60–100 °C; (iii) the warm mix asphalt (WMA) produced at 110–140 °C; and (iv) the hot mix asphalt (HMA) produced at 150–180 °C. Figure 1 shows the classification of the asphalt mixes based on the production temperature [11]. It is noted that the energy required to achieve the desired workability is exponentially increased from the CMA to HMA, resulting in higher GHG emissions of HMA compared to other mixtures.

The primary reason and motivation for adopting WMA techniques are to produce an asphalt mixture at a temperature 10–40 °C lower than the conventional hot mix asphalt (HMA), as shown in Figure 2. The low production temperature of WMA has three benefits. It can significantly reduce environmental burdens, including global warming [2] and the emissions of gasses and fumes [12,13,14,15,16,17,18,19,20,21,22]. The economic benefit of the lower production temperature is directly proportional to the low energy consumption [18,23,24,25,26,27,28,29,30,31,32,33,34], which reduces the financial costs [35,36,37]. WMA production and paving are beneficial because they modulate the mixture viscosity, enhance mixture workability, facilitate compaction [38,39,40], allow the use of reclaimed asphalt pavement (RAP) [41,42,43,44], and provide better working conditions and a healthy work environment [45,46,47]. In more detail, and from the environmental perspective, WMA technology reduces CO_2_ emission based on temperatures during the paving process, which reflect the benefits of paving using WMA techniques that directly affect the workability and compaction of the mixture. WMA techniques serve as compaction aids and minimise the amount of pressure required [38,39,40]. Using the correct laying and compaction temperatures is essential to avoid difficulties. Even though a general temperature drop is permitted within WMA, a little higher temperature between 100 and 150 °C is recommended to be used. The paver screed angle of attack, material movement between the equipment, and thermal segregation could be negatively impacted in certain instances, such as temperature differentials occurring in the surface mix resulted [29]. It is easier to achieve the required densities with WMA in most cases than HMA, even at substantially lower temperatures [27]. This is due to the technologies that have been developed to produce WMA and also to the additives that are used to reduce the viscosity, which makes the mixture easier to manipulate and compact at a lower temperature. However, the operation and maintenance of facilities or plants used for WMA production need additional care to avoid some operational problems [27]. High percentages of RAP can be used in WMA without compromising the asphalt mixture’s workability [48]. Another motivation for using WMA is the possibility of cold weather paving since the mix temperature is closer to the ambient temperature. As a result, the reduction in mixed heat is less dramatic. This closeness of temperatures results in a more extended paving season because there is more time for paving and compaction, and increased hauling distance [35,36,49]. WMA plants can be located close to urban areas because of their low levels of emissions, fumes, and noise. The plants could also be at suitable distances from the construction sites, making it possible to pave in non-attainment areas [24,27,38]. Furthermore, traffic lanes can be opened sooner [50,51,52] due to the small temperature difference reduces the cooling time after construction [38,48]. This is especially desirable in instances such as the rehabilitation of airports and high-traffic roads [20,52,53].

From the literature review, it was revealed that several researchers have studied and compared the possibility of using WMA technology as an alternative to conventional HMA from the physical, rheological, mechanical and performance perspectives. However, reviews on the environmental and cost-effectiveness utilisation of WMA as a sustainable alternative to HMA are very limited. Therefore, this paper provides a detailed and comparative overview of using WMA as a sustainable alternative to conventional HMA from environmental and economic perspectives. The motivations, challenges and research gaps (recommendations and future directions) associated with using WMA technology as an alternative to HMA in asphalt pavement construction are also explored and highlighted in this review to better understand and promote WMA technology for sustainable construction.

## 2. Sustainable Materials

Sustainable development requires using fewer raw natural materials due to the high cost and energy consumption for extraction and transportation. Sustainable development also reduces the emissions of greenhouse gases and uses recyclable materials without compromising the standard requirements. Figure 3 shows that sustainable development comprises three interrelated areas, economic development, social development, and preservation of the environment. In detail, the economic aspect contributes to profits and cost-effectiveness, while the social aspect represents the contribution of the standard of living and equal opportunity to sustainability. Besides, the environmental aspect reflects the natural resources, pollution prevention, and biodiversity. Figure 3 also clearly implied that WMA technology supposes to be consistent with sustainable development that considers the environmental, economic, and social aspects toward equitability and viability [28].

Among the goals of constructing sustainable roads are to ensure safe, comfortable, cost-effective travel, reduce waste generation, and reduce the use of raw materials. It prevents the plundering the natural resources by using waste materials as a substitute [54]. Using waste materials in road construction can reduce the overall environmental impacts [55] and requires developing energy-efficient and eco-friendly paving technology [56,57]. Warm mix asphalt technologies enable the utilisation of higher percentages of recycled materials [28], which facilitate the design of perpetual and sustainable pavement based on the 4R policies (reclaim, recycle, reuse, and reduce). Using waste materials in pavement construction and rehabilitation can reduce energy consumption. Various types of recycled aggregates are used in WMA pavements, including reclaimed asphalt pavement (RAP), Recycled Asphalt Shingles (RAS), construction and demolition, and industry by-products (for example, copper or steel slags) [58]. The primary advantage of WMA is the potential to use a higher quantity of RAP [13,21,37,41,42,43,44,59,60,61]. Using RAP to replace the raw material eliminates the need to extract base raw materials and dump asphalt; this reduces the material and end-of-life consequences [31].

Table 1 shows the effect of using different recycled materials in WMA pavements. It can be reported that many recycled materials that were used as a partial or total aggregate replacement led to an improvement in the mechanical performance of WMA mixtures. The addition of RAP materials to the WMA mixture results in a reduction of permanent deformation due to the enhancement of the stiffness modulus of the RAP/WMA-modified mixtures [62]. Furthermore, the composite of RAP and WMA technologies led to improved fatigue resistance mixtures as a result of the balance between the stiff RAP materials and WMA additives that reduce the viscosity and stiffness of the asphalt [63]. Steel slag and furnace slag as waste materials showed an improvement in the fatigue resistance of asphalt mixtures due to the enhanced stiffness modulus [64,65]. It can also be noticed that the combination of RAP materials with steel slag, crumb rubber or glass fibre results in better moisture, fatigue and rutting resistances [63,66,67]. In contrast, using high RAP materials content in asphalt mixture led to lower moisture susceptibility and fatigue resistances [62,68]. As a fibre additive to WMA, jute fibre significantly improves fatigue and fracture resistance due to the enhancing of the adhesion properties of aggregate and binders toward adequate tensile strength [69]. It was also claimed that the addition of hydrated lime and nano-hydrated lime to WMA as fillers enhances the moisture damage resistance as a result of improved cohesion and adhesion properties [70,71].

## 3. Components and Production of Asphalt Mixture

WMA and HMA have the same components. WMA is easy to use, and its production does not require major modifications to the existing HMA plant. However, the manufacturing of HMA contributes to a higher percentage of CO_2_ emissions both in the initial construction stage [65] and the rehabilitation process [66]. The only difference between WMA and HMA is the production temperature [74]. The preparation of HMA requires a high-temperature range of 150 to 180 °C, while the WMA is prepared at a temperature range of 110 to 140 °C [48,75,76,77,78,79,80]. Table 2 presents the advantages and disadvantages of HMA and WMA [21,31,48,77,81,82].

The mixing and compaction temperatures of WMA can be reduced using organic additives, chemical additives, and water-foaming techniques [9,83,84]. In 2022, Rahmad et al. investigated the use of PG76 in integration with a chemical WMA additive to reduce the temperature during compaction based on environmental sustainability aspects, Rediset, and groundwater and soil contamination. However, it was found that there had been no chemical reaction between PG76 and Rediset. It was also found that after 64 days submerged under water, Rediset-PG76 had no effect on the adjacent water source and soil [8] summarises the different additives and technologies for WMA [21,31,58,84,85]. Even though the technologies differ, they all seek to reduce bitumen viscosity, enhance workability, reduce emissions, and maintain the desired performance. Several studies have shown in Table 3 that these technologies can reduce air pollutants (emissions) and energy consumption [12,21,22,27,49,58,81,83,86,87,88,89,90,91]. Even though the low temperature for producing the mixes the production and paving has several advantages, it could result in poor performance, such as incomplete aggregate drying, poor bitumen coating, and moisture susceptibility due to the presence of water. However, researchers have conducted extensive investigations on these issues and proposed solutions [21]. Furthermore, detailed studies on the cohesion and adhesion failure mechanisms based on advanced laboratory techniques and computational simulation could help in further understanding the reasons behind such common issues toward proposing solutions. In addition, a composite of polymers and nanomaterials into WMA technology could mitigate such moisture susceptibility problems.

Generally, organic additives such as wax or fatty amides reduce asphalt binder viscosity at temperatures over their melting point. These additives should have a melting point higher than the maximum service temperature of the asphalt mixture to increase the rut resistance of the asphalt at high temperatures and limit embrittlement at low temperatures [28]. Chemical additives are liquid surfactants that act at the microscopic interface and do not change the asphalt binder’s viscosity; they are surface agents that increase wetting qualities by lowering the tension between asphalt binders and aggregates and thus reduce internal friction [92,93]. Foaming technologies lower the asphalt binder viscosity by introducing small amounts of water into the hot asphalt binder. As the water evaporates, it expands the binder and reduces binder viscosity; this results in a better aggregate coating. The degree of expansion is dependent on several factors, such as binder temperature and water content [52].

An asphalt mixture is a composite of aggregates, asphalt binders, and fillers. Additives or modifiers are occasionally added to the asphalt binder to improve its performance [94,95,96,97,98]. Aggregates are the main element of asphalt pavements and constitute almost 95% of the mixture. The high percentage of aggregates in asphalt pavements has increased the demand for aggregates in road construction applications. The aggregate materials are often used for the lower pavement layers, such as the base or subbase layer. In 2015, 2660 million tons of aggregates were produced in Europe from quarries, with the UK contributing 110 million tons per year. In addition, France produces approximately 250 million tons per year [99,100]. Malaysia produced 118 million tons of aggregates in 2011 and 160 million tons in 2015 [99,100]. In the United States, aggregate production increased from 1.34 billion tons in 2015 to 1.53 billion tons in 2019. About 72% of the aggregates were used as construction aggregate, primarily for road construction [101].

There are two main phases in asphalt pavement construction and the production and construction of asphalt mixture. The first phase consists of aggregate stacking, heating the aggregates and asphalt binder, and mixing. The second phase is transporting, paving, and compacting the asphalt mixture. The energy consumption during asphalt mixture production is considerably higher than in the transportation and construction phase [102]. The production stage involves heating the aggregates and asphalt and mixing the asphalt mixture. The aggregate heating process for HMA contributes to 67% or more of the total carbon emission, while the asphalt heating and mixing processes contribute only 14% and 12%, respectively [89,103]. According to Stotko [104], about 60% of the energy consumption at the asphalt plant is for drying the aggregates. Peng, Tong, Cao, Li and Xu stated that 76.41% of the total carbon emission is during aggregate heating, while asphalt heating emits 15.67% of the carbon [103]. The moisture content of the aggregate is one of the factors determining the amount of energy consumed during the aggregates drying process [5,76,105,106].

Moreover, the specific heat capacity of the aggregate materials is a critical determiner of the fuel needs and CO_2_ emissions of WMA and HMA. The same type of aggregate extracted from different sources may have different specific heat capacities even if their specific gravities are similar [107]. Jamshidi et al. [108] investigated the effects of the thermal properties (specific heat capacity) of asphalt binders and aggregate materials on energy consumption and environmental footprints of HMA and WMA. The results showed that using low-specific heat capacity aggregates is more energy-efficient and environmentally friendly. The difference in energy requirements varies with the moisture content [104]; a 1% increase in moisture content results in a 3.5% higher energy consumption to dry the aggregates [106]. Another study has shown that energy consumption increased by 1% for every 0.7 L moisture content [109] and that one of the ways to reduce energy consumption is by reducing the mixing temperature [110]. The energy demand is about 2.62 kWh for a 10 °C increase in the mixture temperature and 8.21 kWh for every 1% increase in moisture content [105]. The fuel for heating or drying the aggregates is one of the sources of emissions, where the energy consumption and CO_2_ differ with the type of fuel, as shown in Table 4. It can be seen that using natural gas to heat the aggregate results in the lowest CO_2_ emission compared to different fuel types reported in Table 4, however, natural gas showed to be the highest heating energy required. On the other hand, using fuel oil (N°1/2) as a heating energy source showed to be the lowest among all fuel resources, with a reduction of 9.45% compared to required natural gas energy. However, using fuel oil (N°1/2) results in about 480% CO_2_ emission higher than the emission due to using natural gas. Furthermore, it was reported that the use of natural gas instead of heavy oil to heat the aggregates reduces carbon emissions by 27.72% and the cost by 18.63% [89]. According to Stotko [104], using WMA could reduce fuel oil consumption by about 8400 GJ and prevent CO_2_ emission by 620 tons annually based on an asphalt plant in South Africa.

## 4. Life Cycle Assessment

The aspects affecting the several phases of the pavement life cycle at various levels for achieving scientific, reasonable calculations of energy consumption and carbon dioxide emissions over the pavement life cycle development aims to balance environmental, economic, social, and political goals to save the earth for future generations [111,112,113,114]. Thus, an environmentally friendly grading system is necessary to measure the environmental effect of asphalt pavements. Häkkinen and Mäkelä [115] introduced the life cycle assessment (LCA) of asphalt pavements in the mid-1990s. Figure 4 shows the main stages of LCA: (a) extraction and processing of raw material, (b) transportation, (c) construction, (d) utilisation, (e) maintenance and repair, and (f) final disposal at the end of life [13,27,116,117,118,119,120,121,122,123,124,125]. Each year, the production of asphaltic mixture consumes a massive amount of energy and emits CO_2_. Besides, the challenge to achieve reductions in asphalt pavement production should focus on the industrial stages of content materials and the producing process of asphalt mixtures. While this is the case, the raw materials also contain minerals, and the use of such materials for industrial production may be regulated by the environmental threshold values [126]. Asphalt pavements have significant environmental burdens, including WMA and HMA production with various variables (e.g., aggregate and binder) that release emissions during their life cycle starting from the plant, construction site, and long-term exposure to climatic conditions [2].

Researchers have conducted experimental studies on the LCA of WMA [2,7,8,12,13,16,27,127,128,129,130,131,132]. Figure 5 shows the result of the LCA analysis of the environmental impact assessment of WMA [7,8]. The figure shows a 24%, 18%, 10%, and 3% reduction in the environmental impacts of air pollutants, fossil fuel depletion, smog formation, and global warming, indicating that, overall, WMA has 15% less environmental impact than HMA [2].

Cheng, Chen, Yan and Zheng [127] performed an LCA on WMA and HMA and found that using WMA could reduce photochemical ozone formation (POF) and fuel utilisation by 65–75% and 20–25%, respectively. The cradle-to-grave analysis performed by Blankendaal et al. [133] showed that using WMA instead of HMA mixtures reduced energy consumption by 33%, leading to fewer emissions. Wu and Qian [131] used the life cycle assessment to compare WMA that was prepared with chemical agents with HMA. They observed that the environmental impact of the chemical agent based-WMA mixture is less severe than the conventional HMA mixture due to the lower manufacturing temperature required for WMA production.

Tatari, Nazzal and Kucukvar [129] used a hybrid LCA model to compare the environmental benefits of the mixtures prepared with different types of warm mix additives. The result exposed that Sasobit and Evotherm-modified asphalt mixtures emit minor pollutants and considered warm mix additives using Evotherm and Sasobit to reduce production temperatures. Hence, the researchers found that Rediset improves safety and sustainability and protects environmental health [134]. Ma, Sha, Lin, Huang and Wang [12] compared the life cycle assessment for WMA and HMA pavements and found that the WMA pavements emit less CO_2_ during their life cycle and thus are more environmentally friendly. The extraction of raw materials used in the construction of WMA and HMA pavements has a significant impact on the environment. Vidal, Moliner, Martínez and Rubio [13] evaluated the environmental impacts of reclaimed asphalt pavement with zeolite-based WMA and HMA. The result showed that zeolite-based WMA pavements have similar impacts as HMA pavements with the same reclaimed asphalt pavement (RAP) content during the entire life cycle.

## 5. Energy Consumption and Economic Benefits

Modern civilisation must address several critical issues to achieve sustainable development. There is an urgent need to reduce energy consumption to reduce climate change; it is also essential to reduce raw materials’ utilisation to reduce waste [135]. The considerable amount of energy used by the asphalt pavement industry has an adverse impact on the environment. The materials, plants, and machinery used at asphalt pavement construction sites have an adverse impact on the environment through the generated wastes, discharged water, and emissions. Gillespie [136] used regression analysis to predict the amount of energy required to produce asphalt mixtures. The result showed that the process consumed approximately 9 L/ton of fuel and 8 kW/ton of electricity and emitted 28.8 kg/t CO_2_.

The lower energy consumption in WMA production is a significant economic benefit in pavement construction. According to Kristjánsdóttir et al. [137], WMA is especially beneficial in areas where fuel prices are high. The lower production temperature in WMA production is directly proportional to the reduced energy consumption [23,138]. Based on the varying temperature reduction ranges, a comparison of WMA and HMA revealed that the warm technology could reduce energy consumption by 20% to 75% [2,18,24,25,26,27,28,29,30,31,32,33,34]. This very wide range in the reduction of energy consumption could be attributed to the different WMA technologies adopted by different studies and the combination of WMA with other technologies, such as RAP technology. Furthermore, the reduction in energy consumption is also dependent on the fuel type and energy source [9]. The energy consumption in asphalt production varies depending on the country and region [136]. Hence, compared to HMA, the reduced energy consumption associated with WMA production resulted in 12–14% fuel savings and an average energy cost savings of $1.61 per ton of mixture in Louisiana state, USA [2]. One benefit of using RAP is the minimal maintenance and rehabilitation costs and environmental impact [139]. Moreover, it is possible to significantly reduce the amount of asphalt binder used in pavement construction. For example, using 100% RAP in HMA can reduce the construction cost by 79.7% compared to the mixture without RAP [95]. In WMA, adding 15% RAP reduces all endpoint consequences by 13–14%, including climate change, fossil depletion, and total cumulative energy consumption [13]. Moreover, using 30% RAP and 0.3% natural zeolite has a considerable cost-saving benefit, which reduces cost by about 25% compared to HMA [41]. According to Almeida-Costa and Benta [76], depending on the type of asphalt mixtures, the energy consumption for producing HMA and WMA differs by 8.6–18.4%. Oner and Sengoz [37] analysed the cost-benefit of HMA and WMA without and with varying percentages of RAP (10, 20, and 30% of the total mix weight) when using different warm additives. The result showed that using 30% RAP with organic additive is the most economical in terms of the final cost in Turkish lira (TL) for all distances from the plant to the construction site, as shown in Figure 6 [37].

The production stage of WMA at 120 °C brought 24,831 gigajoules of energy savings for 140,000 tonnes every year due to adding Ca(OH)2-incorporated zeolite [90]. Romier et al. [140] investigated the drying and heating processes in the production of HMA and WMA; the heat balance of HMA is 175 MJ, and 83MJ for WMA, which is a 50% reduction in the heating energy per tonne of WMA. Oliveira, Silva, Fonseca, Kim, Hwang, Pyun and Lee [14] examined the fuel consumption for producing WMA and HMA mixtures to determine plant efficiency. The production of HMA consumes 9.3 L/ton of fuel, and WMA production consumes 6.3 L/ton, which is 32% less fuel consumption than HMA.

According to Jain and Singh [77], the fuel consumption in HMA production is (6.2–7.2) kg/ton and (5–6) kg/ton for WMA. Hettiarachchi et al. [141] have shown that reducing the production temperature of the asphalt mixture by 20 °C reduced the energy consumption by 25%. Likewise, Prowell, Hurley and Frank [29] stated that it is possible to reduce the fuel consumption in WMA production by 30–35%. They estimated that lowering the production temperature by 6 °C could reduce fuel usage by 3%. Moreover, theoretical estimates show that lowering the temperature by 28 °C results in an 11% savings on petroleum fuel [142]. According to Hassan [8], compared to HMA, WMA uses 18% less fossil fuel. However, the cost of warm mix additives increases the cost of producing WMA. [143] stated that WMA could increase the cost of asphalt mixtures between $2 to $4 per tonne of the mix. In terms of cost analysis, HMA and WMA were initially compared based on including organic, chemical, and foaming WMA additives in terms of materials, mix heating and transportation costs. The results showed that organic and foaming could reduce costs slightly when compared to HMA. Meanwhile, chemical additives may slightly raise the cost when compared to HMA. Moreover, the same study concluded that adding RAP could significantly decrease the cost of WMA production in comparison with HMA [141]. Additionally, the financial advantages from energy savings could outweigh the expenses of WMA additives and machine installation [21,144]. In their analysis of seven plants, Bueche and Dumont [106] found that the average energy consumption for HMA production is 356 MJ/t and 226 MJ/t for WMA. The lower temperature in WMA production may also result in additional cost savings since the asphalt plant undergoes less wear and tear [2].

## 6. Greenhouse Gas (GHG) Emission

The carbon footprint measurement covers two main processes. Off-site activities are the production and transportation of materials, for example, coarse aggregate and fine aggregate, fillers, and asphalt binders. Onsite activities are the plant operation for producing asphalt mixtures and laying the sub-base, base, and surface courses [12,103,145].

Intergovernmental Panel on Climate Change [146] reported that the major greenhouse gases are carbon dioxide (CO_2_), methane (CH_4_), nitrous oxide (N_2_O), hydrofluorocarbons (HFCs), perfluorocarbons (PHCs), and sulfur hexafluoride (SF_6_) CO_2_, CH_4_, and N_2_O have a substantial impact on human activities and considerable greenhouse effects [12,147,148]. According to the global warming potential (GWP) proposed by the Intergovernmental Panel on Climate Change [146], as shown in Table 5, the different greenhouse gases can be converted into their CO_2_ equivalent emission, the GWP of CO_2_, shown in Table 6.

The carbon account of asphalt pavement is the sum of all relevant emission sources. The total sum of the asphalt pavement carbon footprint is expressed in Equation (1) as follows.
(1)$$\overset{n}{\underset{i=1}{\sum }}{\left(CO_{2}e\right)}_{i}=\overset{n}{\underset{i=1}{\sum }}(AD_{i}x\;EF_{i}x\;Gwp_{i})$$
where *CO*_2_*e* is the carbon equivalent emission from a single procedure in asphalt pavement, *AD* is the activity data, *EF* is the carbon emission factor, and *GWP* is the global warming potential.

At present, researchers of innovative asphalt material technology focus on green asphalt mixtures because the carbon energy consumption of these materials is several times lower than for asphalt production. In the future, asphalt pavements must be considered a part of a symbiotic framework between buildings and nature. However, flexible pavement has primary constituent materials: asphalt and coarse aggregate. Hence, asphalt production has a long process of petroleum distillation residue and has an emission factor of 11.91 kg CO_2_/gal [145]. Figure 7 shows the total greenhouse gas (GHG) emissions in asphalt pavement construction [12]. The critical phases are the raw materials production and asphalt mixing phases, which contribute 97.19% of the total GHG emissions, of which 43.18% is from raw materials production and 54.01% from asphalt mixing. The transportation of raw materials and asphalt mixture contributes 1.35% of the total GHG emissions, while the laying and compacting phases generate only 0.86% and 0.61% of the total GHG emissions [12]. It is essential to minimise the carbon footprint of asphalt materials to meet the global target of reducing GHG emissions. The Climate Change Act (2008) targets reducing 80% of GHG emissions by 2050 based on the 1990 baseline [90,149]. In 2020, the primary reasons for the 5.8% record-breaking increase in global CO_2_ emissions to an absolute maximum of 33.0 billion tons were the continued growth in developing nations and economic recovery in the industrialised countries [150]. Therefore, novel technologies in asphalt pavement, such as WMA, could play a role in reducing GHG emissions.

The low production temperature of WMA contributes directly to reducing GHG emissions. Several studies have shown that WMA pavements emitted less GHG than HMA [12,13,14,15,16,17,18,19,20,21,22,144,151,152]. Table 7 shows the percentage of reduction in GHG emissions when using warm mix technology compared to conventional HMA technology. Compared to various WMA additives used, as shown in Table 7, it can be seen that the Evotherm resulted in the highest CO_2_ reduction of 17 to 60%, followed by the foaming technology of 58.5% reduction. On the other hand, double barrel green exhibited the lowest reduction of 10.9%, followed by synthetic zeolites with a 15.5% reduction. These indicate the out-performing of Evotherm additive in the CO_2_ reduction compared to most of the other WMA additives and processes reported. In terms of CO reduction, it is clear that the foaming technique, Sasobit, Evotherm and Aspha-min, showed the highest reductions of 91.9, 63.2 and 63 and 62%, respectively. Similar to CO_2_ reduction, double barrel green showed the lowest reduction of 10.4% in CO. Similar to CO reduction, the foaming technique exhibited the highest reduction of SO_2_ at 99.9%, followed by Sasobit and Aspha-min of 83.3% reduction for each. However, in the case of SO_2_ reduction, synthetic zeolites led to the lowest reduction among all reported additives and techniques. Evotherm and double barrel green revealed the highest and lowest NO_x_ reduction among all WMA additives and techniques reported in Table 7, with 72.6% and 8.3%, respectively. Based on the aforementioned discussion, it can be stated that Evotherm and foaming techniques outperformed the other WMA techniques in terms of emission reduction, including CO_2_, CO, SO_2_ and NO_x_. In contrast, Aspha-min and Sasobit resulted in a significant reduction in VOC.

The main greenhouse gases emitted in road construction are carbon dioxide (CO_2_), nitrous oxide (N_2_O), and methane (CH_4_). However, these gases do not contribute equally to polluting the atmosphere since the emission of CO_2_ is considerably higher than other gasses [155]. According to Keches and LeBlanc [151], using WMA instead of HMA can prevent the emission of 3,774,000 tonnes of CO_2_, which is a 43.9% reduction in CO_2_ emission.

## 7. Health Hazards for Workers

Asphalt is the non-distillable component of crude oil. This extremely viscous substance traps small quantities of volatile and semi-volatile chemical molecules [156]. Heating the asphalt over the softening point and agitating it releases pollutants, thus exposing the workers to the pollutant [157]. The low gas emissions in WMA technologies improve working conditions [31,46,47] and reduce the workers’ exposure to respirable fumes and gases released during the asphalt paving process [24]. Furthermore, replacing the conventional HMA with WMA has considerable health benefits, provides a healthy work environment, and reduces exposure to occupational risks [45]. The low mixing temperature ensures a more comfortable working environment, which could help in worker retention [158]. Several studies have shown that workers exposed to asphalt fume have a higher risk of cancer [159,160,161,162,163]. In 2013, the International Agency for Research on Cancer, affiliated with the World Health Organization, classified occupational exposure to straight-run bitumen and its emissions during road paving as “possibly carcinogenic to humans” (Group 2B). [164], Fuhst et al. [165] conducted inhaling research and exposed Wistar rats to asphalt fume for 2 years. They concluded that asphalt fume is not tumorigenic to rats when inhaled. However, they detected asphalt-related irritating effects in the nasal passages and lungs of the rats.

Workers in the road construction sector are also exposed to other health problems. There is evidence of possible sub-chronic irritative inflammatory effects in the lower airways of the respiratory system for workers exposed to asphalt [166]. Tepper et al. [167] demonstrated statistically significant throat symptoms among workers exposed to fumes. The workers often experience health symptoms such as fatigue, reduced appetite, eye irritation, and laryngeal-pharyngeal irritation [168].

## 8. Limitations, Recommendations and Future Directions

Even though there are many advantages to using WMA technology in pavement construction, the studies showed that there are limitations. One of these limitations is the limited durability of WMA, which may not fulfil the requirements for very heavy and extreme traffic, such as airport pavement. Such limited durability was justified due to the short and long-term performance of WMA technologies against the mechanical and environmental conditions of traffic loading and high-temperature weather [28,169,170]. Another limitation is the propensity of WMA for water damage or stripping due to the low mixing and compaction temperatures may reduce the aggregate and binder adhesion [28,169,170,171]. How to look for WMA additives alternatives with as much waste materials as possible and based on biomaterials technologies toward further sustainable and environmentally friendly WMA technologies is also considered another challenge facing the researchers and the pavement industry [171]. The introduction of RAP materials into the WMA technology still faces challenges, such as the oxidised RAP materials that adversely affect the adhesion and cohesion mechanism of the aggregate-binder interface system [37]. Therefore, a combination of WMA and RAP technologies in one product that is durable, cost-effective, and environmentally friendly needs further studies. Furthermore, maximising the possibility of recycling WMA to-ward 100% recycled materials with mitigating their mechanical and environmental issues is another challenge that needs to be considered.

In this section, the most important recommendations and future directions to further develop a sustainable WMA technology in terms of environmentally friendly, safe, and cost-effective are also summarised. These recommendations may help in providing a useful reference for other researchers and the pavement industry interested in developing a sustainable alternative to conventional asphalt technologies.
The composite of WMA technology with different asphalt technologies such as RAP and bio-asphalt technologies are strongly recommended to be further studied in order to mitigate the environmental and CO_2_ emissions and energy consumption of conventional technologies.Advanced optimisation, modelling, and simulation methods such as machine learning are also recommended to be applied with respect to studying the environmental and energy consumption of WMA technology separately and combined with other relative technologies.Investigating the possibility of using waste materials in WMA technologies and comparing their environmental and economic impacts to the common conventional WMA additives.The long-term environmental impact of different additives that are used as WMA additives is another research aspect that should be studied.Validating the laboratory findings on the environmental and economic benefits of WMA technology that have been reported in the literature by conducting field studies over the different regions in various environmental conditions.Standards and specifications that are needed to guide researchers and pavement industries in using WMA technology in a wide range of developed and developing countries still need to be established.

## 9. Conclusions

This paper has presented a comparative overview of the WMA and HMA used to construct road pavements from environmental and economic perspectives. The higher demand for road construction in recent decades has also raised the issues of the negative impacts of road infrastructure on the ecosystem. As a result, there are efforts to reduce these adverse impacts of road construction. According to the current review of the literature, the following conclusions can be drawn:Generally, asphalt mixture production comprises aggregate heating, asphalt heating, and asphalt mixing. However, the highest percentage of energy in the asphalt mixture production and afterwards the carbon emission occurs during aggregate heating;The energy consumption and emissions in the production of asphalt mixtures are related to many factors, such as the type of aggregate and its heat capacity, aggregate moisture content, type of fuel, fuel consumption, and production temperature;Depending on the asphalt mixture, the production temperature of WMA is 10–40 °C lower than the conventional hot mix asphalt (HMA). This reduction in temperature positively affects several aspects, such as fuel consumption and CO_2_ emissions;Furthermore, in terms of environmental benefits, the use of WMA can reduce the emission of gases and fumes and global warming. Concerning the LCA of WMA compared to HMA in terms of environmental aspects, it is inclined to be more favourable for WMA;The economic benefit of warm technology is the reduced financial cost because WMA uses 20–70% less energy;The low production temperature also causes less wear and tear to the plant and thus provides additional cost savings;The paving and working conditions, organic additives, and foaming technologies reduce asphalt binder viscosity and thus enhance workability and facilitate compaction;Moreover, low viscosity availability encourages an increase in reclaimed asphalt pavement content and, therefore, provides lower application temperatures;In addition, the benefits are coupled with the application of RAP into the new asphalt binder involving WMA additives;The other benefits of the paving temperature being closer to the ambient temperature and heat are less dramatic: longer hauling distance, sooner opening of a traffic lane, less exposure to fumes, and a healthy work environment, which reduces the risk of health problems among the workers.

## Figures and Tables

**Figure 1 ijerph-19-14863-f001:**
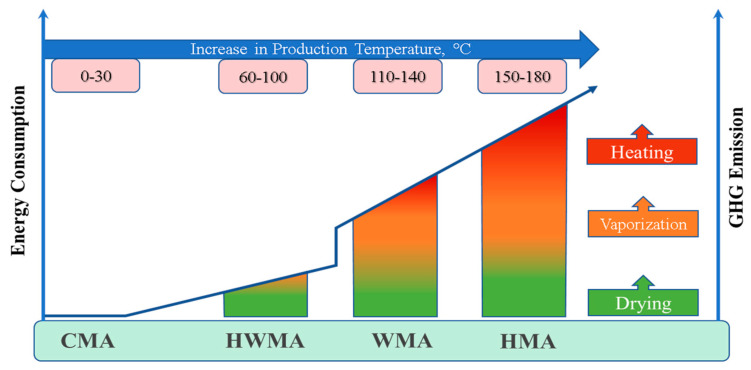
Classification of asphalt mixes based on production temperature [11].

**Figure 2 ijerph-19-14863-f002:**
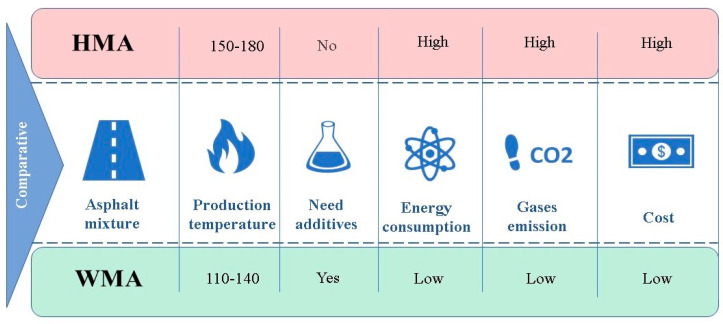
Comparison of HMA and WMA.

**Figure 3 ijerph-19-14863-f003:**
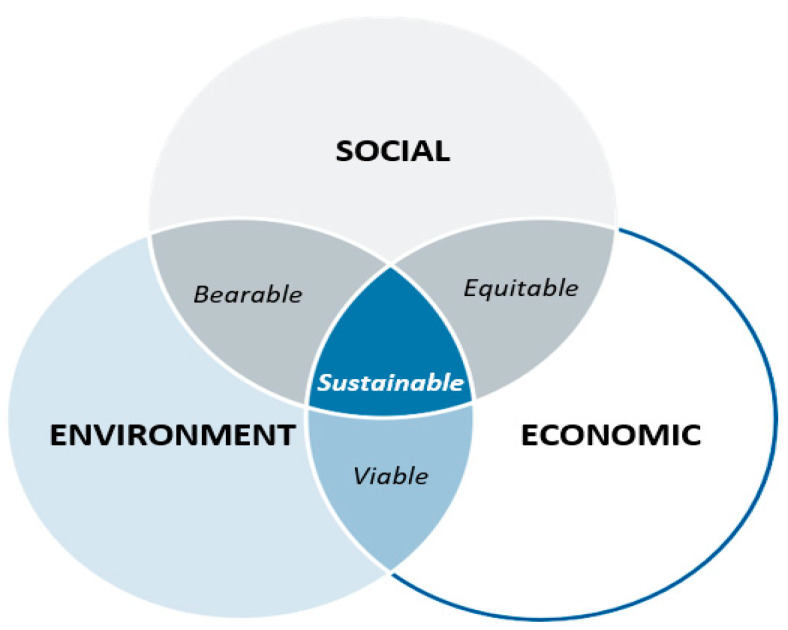
Sustainable development [28].

**Figure 4 ijerph-19-14863-f004:**
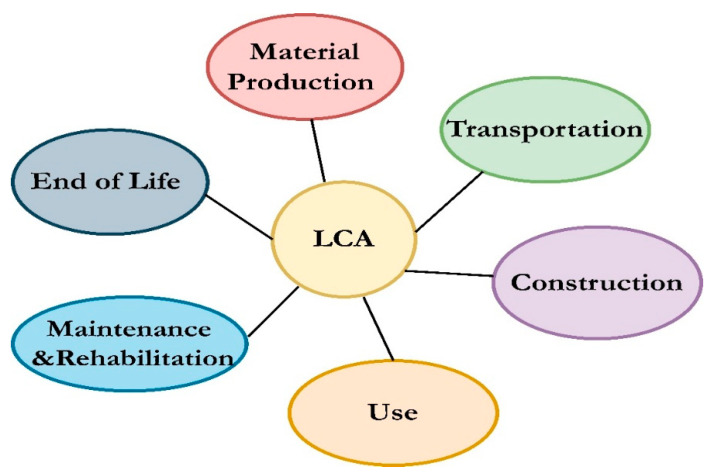
The LCA stages for asphalt pavement.

**Figure 5 ijerph-19-14863-f005:**
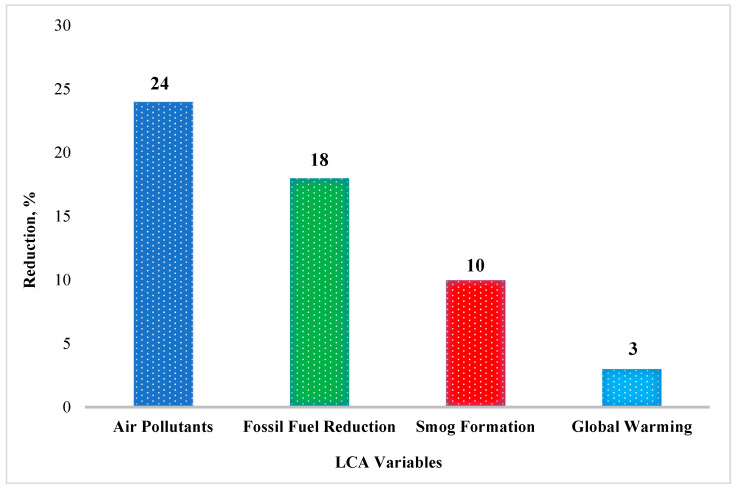
Reduction in different LCA categories.

**Figure 6 ijerph-19-14863-f006:**
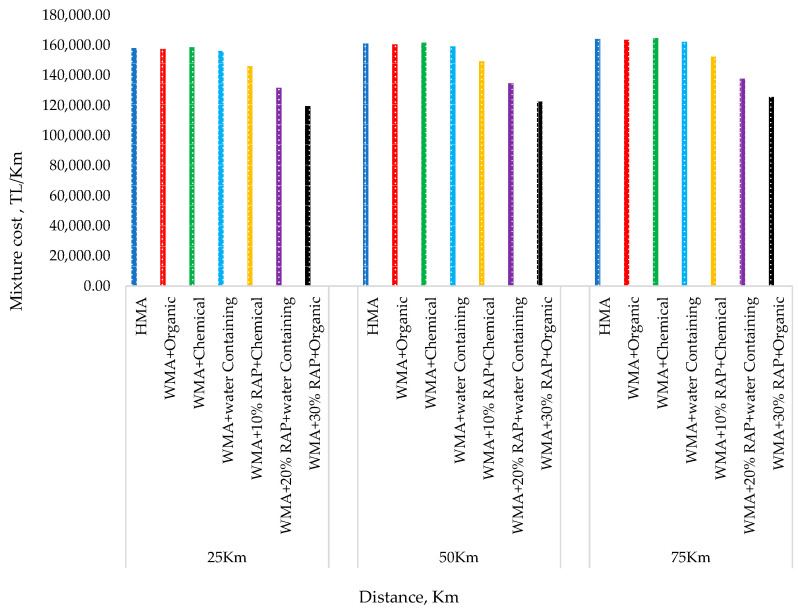
Cost analysis for HMA and WMA with RAP.

**Figure 7 ijerph-19-14863-f007:**
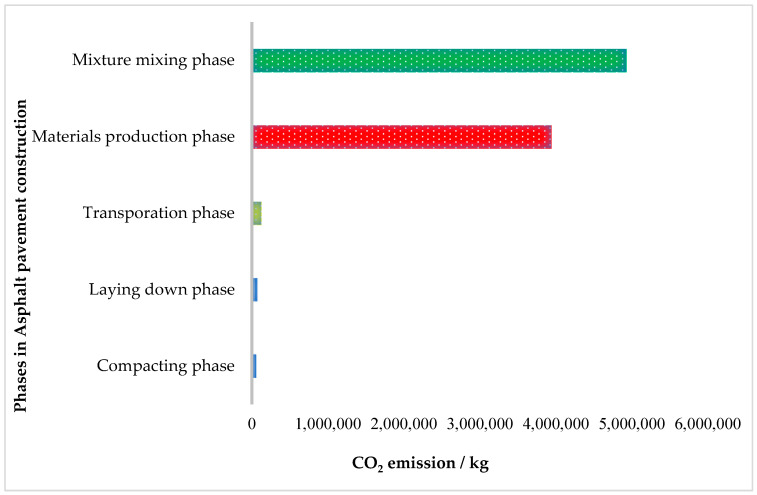
CO_2_ emission in asphalt pavement construction.

**Table 1 ijerph-19-14863-t001:** The recycled materials used in WMA.

Usage	Type of Waste and WMA Additive	Effect	Reference
Aggregate	Glass (10% *) + Zycotherm^TM^ (0.05, 0.10, 0.15 and 0.20% **)	Reduce resilient modulus, creep, and moisture susceptibility	[72]
	Furnace slag (30% *) + Sasobit^®^ (4% **) or Rediset^TM^ (2% **)	Improve fatigue resistance and stiffness modulus	[64]
	Steel slag (40% *) + Surfactant-based chemical additive (0.5% **)	Improve the fatigue resistance and mechanical properties of asphalt mixtures	[65]
	RAP (0, 20 and 40% *)+ Steel slag (0 and 40% *) + Sasobit^®^ (1.5% **)	RAP improves moisture sensitivity and resilient modulus Steel slag improves the resilient modulus The mixes containing RAP and/or slag have a lower rutting potential The WMA containing RAP and/or steel slag has enhanced fatigue resistance	[63]
	RAP (30 and 60% *) + crumb rubber (CR) (0, 10 and 20% *) + Sasobit^®^ (4 and 5.5% **)	RAP and crumb rubber have a positive effect on moisture susceptibility The result of the fatigue test showed that using RAP and CR improves the fatigue resistance of the asphalt mixtures	[66]
	RAP (0, 20, 40 and 50% *) + Glass fibre (0.3% ***) + Sasobit^®^ (1.5% **)	Improved rutting and moisture susceptibility resistance	[67]
	RAP (20, 30, 40, 50 and 60% *) + Mobile engine oil (10, 12.5, 15, 17.5 and 20% **) + Evotherm^TM^ (0.5% **)	Higher RAP proportion results in lower OBC of the RAP-WMA mixes The tensile strength ratio (TSR) decreased with higher amounts of RAP material. Higher rejuvenator dosage reduced the TSR	[73]
	RAP	The use of WMA increases permanent deformation but adding RAP in the mixture resulted in less rutting	[62]
	RAP	The 50% RAP WMA has a good fatigue performance	[68]
Fibre	Jute fibre (0, 0.3, 0.5 and 0.7% ***) + Sasobit^®^ (3% **)	Enhanced fracture resistance	[69]
Additive or filler	Hydrated Lime (1% ***) + Advera (0.25%**), Sasobit (3.0%**), and Cecabase RT (0.35% **)	Enhanced moisture susceptibility	[70]
	Nano hydrated lime (1% ***) + Aspha-Min (0.3% ***), Evotherm (0.5% **), and Sasobit (1.5% **)	Increase the indirect tensile strength (ITS) and TSR	[71]

* By aggregate weight, ** By asphalt binder weight, *** by mixture weight.

**Table 2 ijerph-19-14863-t002:** The advantages and disadvantages of asphalt mixes.

Mix Type	Production Temperature	Advantages	Disadvantages
Hot-mix asphalt	150–180 °C	Superior mixture performance Lower initial cost	High production temperature High emissions High energy consumption
Warm-mix asphalt	110–140 °C	Low production temperature Low emissions Energy saving Better working conditions Longer haul distance Minor wear and tear on the plant Less binder ageing	Low mixture performance Higher initial cost due to the use of additives Poor aggregate coating and bonding

**Table 3 ijerph-19-14863-t003:** Warm mix technologies and additives.

Type of Additive	WMA Process	Product	Company	Dosage	Location	Temperature °C
**Organic Additive**	FT Wax	Sasobit^®^	Sasol	1.0–2.5% *	Worldwide	20–30 (R)
	Montan Wax	Asphaltan B	Romonta GmbH	2.0–4.0% *	Germany	20–30 (R)
	Fatty Acid Amide	Licomont BS	Clariant	3.0% *	Germany	20–30 (R)
	Wax	3E LT or Ecoflex	Colas	Not specified	France	20–30 (R)
**Chemical additive**	Emulsion	Evotherm^®^	MeadWestvaco	0.5–0.7% *	USA, worldwide	85–115 (R)
	Surfactant	Rediset	Akzo Nobel	1.5–2.0% *	USA, Norway	30 (R)
	Surfactant	Cecabase RT	CECA	0.2–0.4% **	USA, Norway	30 R(R)
	Liquid Chemical	Iterlow	IterChimica	0.3–0.5% *	Italy	120 (P)
**Foaming Technique**	Water-containing	Aspha-Min^®^	Eurovia and MHI	0.3% **	Worldwide	20–30 (R)
	Water-containing	Advera^®^	PQ Corporation	0.25% **	USA	10–30 (R)
	Water-based	WAM Foam	KoloVeidekke and Shell Bitumen	2–5% water *	Worldwide	100–200 (P)
	Water-based	Low Energy Asphalt (LEA^®^)	LEA-CO	3% water with fine sand	USA, France, Spain, Italy	60–80 (P)
	Water-based	Low Emission Asphalt	McConaughey Technologies	3% water with fine sand	USA	90 (P)
	Water-based	LT Asphalt	Nynas	0.5–1.0% *	Netherlands	90 (P)
	Water-based	LEAB^®^	Royal Bam Group	0.1% *	Netherlands	90 (P)
	Water-based	Double Barrel Green	Astec	2.0% water *	USA	116–135 (P)

* By asphalt binder weight, ** by mixture weight, P; Production temperature, R; Reduction temperature.

**Table 4 ijerph-19-14863-t004:** Energy and CO_2_ emission by different fuel types.

Fuel	Heating Energy for Aggregate [110]		CO_2_ Emission	
	Value	Unit	Value	Unit
Diesel	42,791,000	J/kg	2.6390	kg/L
Heating oil	42,612,000	J/kg	-	-
Fuel oil (N°1/2)	42,686,000	J/kg	3.2160	kg/t
Natural gas	47,141,000	J/kg	0.1836	kg/kWh
Propane gas	46,296,000	J/kg	-	-
Electricity	3,600,000	J/kWh	0.5410	kg/kWh

**Table 5 ijerph-19-14863-t005:** Carbon emission factor (EF), (mg/MJ).

Gas	Type of Energy				
	Coal	Fuel Oil	Diesel/Petrol	Asphalt	Natural Gas
CO_2_	94,600	77,400	74,100	80,700	56,100
CH_4_	1	3	3	3	1
N_2_O	1.5	0.6	0.6	0.6	0.1

**Table 6 ijerph-19-14863-t006:** Global warming potential.

Greenhouse Gas	CO_2_	CH_4_	N_2_O
**CO_2_ equivalent**	1	21	310

**Table 7 ijerph-19-14863-t007:** The reduction in gas emissions, %.

Reference	Additive or Process	Type of Emission					
		CO_2_	CO	SO_2_	NO_X_	VOC	Dust
Hamzah and Golchin [17]	Rediset	31.7	-	-	-	-	-
Ma, Zhang, Zhao and Wu [16]	Evotherm	60	-	75.2	72.6	-	-
Vidal, Moliner, Martínez and Rubio [13]	Synthetic zeolites	15.8	18.4	9.67	16.5	-	-
Davidson [15]	Evotherm	46	63	41	58	-	-
Oliveira, Silva, Fonseca, Kim, Hwang, Pyun and Lee [14]	LEADCAP	32	18	24	33	-	-
Middleton and Forfylow [18]	Double barrel green	10.9	10.4	−14.3	8.3	-	-
Vaitkus, Čygas, Laurinavičius and Perveneckas [20,53]	-	30–40	10–30	35	60–70	50	20–25
Davidson and Pedlow [152]	Evotherm	17.35	19.51	−17.24	20	-	-
Larsen, Moen, Robertus and Koenders [143]	WAM-foam	31.4	28.5	-	61.5	-	-
Rubio, Moreno, Martínez-Echevarría, Martínez and Vázquez [19]	Foaming	58.5	91.9	99.9	66.7	-	-
d’Angelo, Harm, Bartoszek, Baumgardner, Corrigan, Cowsert, Harman, Jamshidi, Jones and Newcomb [28], Prowell [153]	-	15–40	10–30	20–35	60–70	-	25–55
Sargand, et al. [154]	Aspha-min	-	62	83.3	30.8	62.8	-
Sargand, Nazzal, Al-Rawashdeh and Powers [154]	Sasobit	-	63.2	83.3	21.2	51.3	-

CO_2_: carbon dioxide, CO: carbon monoxide, SO_2_: sulphur dioxide, NOx: nitrogen oxides, VOC: volatile organic compounds.

## Data Availability

All materials used in this manuscript are available upon request to the corresponding author.

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
