# Peer review of "A Comparative Review of Hot and Warm Mix Asphalt Technologies from Environmental and Economic Perspectives: Towards a Sustainable Asphalt Pavement"

_ijerph, 2022, doi:10.3390/ijerph192214863_

Round 1
Reviewer 1 Report
[General Comment] The paper conducts a comparative review between Warm Mix Asphalt (WMA) and Hot Mix Asphalt (HMA) from environmental and economic perspectives. In its current form, the paper contains a lot of redundancy in its different sections. The manuscript needs to be reorganized. The authors tend to mention general terms and sentences rather than elaborating on them, which reduces the paper's readability. The authors disregarded the contribution of the WMA additives in their Life cycle and cost analysis. A section on RAP usage with WMA is suggested. Please consider the following points to improve the quality of the manuscript.
[Comment #1] Please use more recent publications in the literature review section. Here are some related publications to consider.
· Daniela L. Vega Araujo, Joao Santos & Gilberto Martinez-Arguelles (2022): Environmental performance evaluation of warm mix asphalt with recycled concrete aggregate for road pavements, International Journal of Pavement Engineering, DOI: 10.1080/10298436.2022.206499
· Ahmed Abdalla, Ahmed F. Faheem "Life Cycle Assessment of Eco-Friendly Asphalt Pavement Involving Multi-Recycled Materials: A Comparative Study" Journal of Cleaner Production, 2022, 132471, ISSN 0959-6526
· Rodrigo Polo-Mendoza, Rita Peñabaena-Niebles, Filippo Giustozzi, Gilberto Martinez-Arguelles "Eco-friendly design of Warm mix asphalt (WMA) with recycled concrete aggregate (RCA): A case study from a developing country" Construction and Building Materials, Volume 326, 2022, 126890, ISSN 0950-0618
[Comment #2] The research gap is not clear. Many studies have conducted a comparative review of the environmental impacts accompanied by adopting WMA technology. In-Page 1, Lines 29-30, the abstract mentioned that the paper highlights the challenges, motivations, and research gaps in using both HMA and WMA technologies. However, this is not clearly illustrated in the manuscript. What is this paper adding to the current knowledge?
[Comment #3] The acronyms must be defined when stated for the first time in the manuscript, such as "CR, ITS, TSR" in Table 1.
[Comment #4] [Page 2, Line 51] How many million Km of roads is constructed in 2021, and where?
[Comment #5] [Page 2, Lines 61-62] how much is the contribution of road construction to the overall emissions of greenhouse gases?
[Comment #6] [Page 2, Line 69] Please fix the error of referencing the figures and tables in the whole manuscript.
[Comment #7] Where is the discussion of Figure 3?
[Comment #8] [Page 4, Line 115] RAP cannot be considered a raw material. Please rephrase the sentence.
[Comment #9] [Page 4, Table 1] Where is the discussion of Table 1? Please elaborate on the data mentioned in Table 1. For example, what are the percentages used for each waste material? What organic, chemical additives, and water foaming techniques are adopted in these studies? Do these additives influence the WMA performance?
[Comment #10] [Page 5, Line 138] Add "Table 3" to the beginning of the sentence.
[Comment #11] [Page 5, Line 145-146] Please illustrate the proposed solutions for each issue mentioned in the sentence before.
[Comment #12] [Page 6, Line 166-168] Please add more recent data.
[Comment #13] [Page 6, Line 161-170] What are the authors trying to build by this paragraph in this section?
[Comment #14] [Page 6, Line 177-178] Please confirm these contribution percentages by more than one reference. For what type of mix were these contribution percentages calculated?
[Comment #15] [Page 7, Line 201-203] "According to Stotko [94], using WMA could reduce fuel oil consumption by about 8400 GJ and prevent CO2 emission by 620 tons annually." Is this Globally or in a specific country?
[Comment #16] [Page 7, Table 4] Where is the discussion of table 4? The heating energy mentioned in the table is for what?
[Comment #17] [Page 7, Lines 220-222] The sentence is hard to comprehend; please rephrase.
[Comment #18] [Page 8, Lines 236-240] The authors tend to use vague general terms like "Negative environmental impacts." Please identify what environmental impacts were measured. Writing ambiguous sentences, like the one in Lines 239-240 and lines 246-248, in literature review papers is inappropriate; please be more specific about the findings.
[Comment #19] [Page 7, Section 4] The authors did not mention the influence of the different Warm mix technologies and additives on the Life Cycle Assessment and cost analysis. This information is essential to be investigated in such a study.
[Comment #20] [Page 9, Lines 268-270] 20% to 75% is a very wide range. Please elaborate on the reason behind this wide range.
[Comment #21] [Page 9, Lines 273-275] in what country and region were these savings observed?
[Comment #22] [Page 9, Lines 273-275] “For example, using 100% RAP can reduce construction cost by 79.7% [87]. Adding 15% RAP reduces all endpoint consequences by 13 14%, including climate change, fossil depletion, and total cumulative energy consumption [10]". Are these results for HMA or WMA, and compared to what?
[Comment #23] [Page 9, Lines 273-275] "[132] stated that WMA could increase the cost of asphalt mixtures between $2 to $4 per tonne of the mix". This cost was analyzed in 2004; please use more recent studies.
[Comment #24] [Page 6, Lines 177- 204] Why did the authors not add these paragraphs and table 1 in section 5, "Energy consumption and economic benefits." The paper needs further organization and avoiding redundancy to increase the paper's readability.
[Comment #25] Remove the dot in line 324 between (SF6) and CO2.
[Comment #26] [Pages 11-12, section 6] how are the paragraphs from lines 318 to 361 related to the research topic? What is the point of Tables 5 and 6, and where is the discussion of these tables? Figure 7 is for HMA or WMA? This section mentioned very general information. It needs more in-depth discussion and comparison between the GHG amounts emitted during the HMA and WMA production and mixing process. Also, authors should not disregard the GHG emissions from different warm mix technologies and additives. The sentence in lines 364-366 is redundant and almost mentioned on every page in this manuscript. The authors shall remove all the redundant sentences.
[Comment #27] [Pages 12, Table 7] Where is the discussion on the data illustrated in table 7. In table 7, the reduction in gas emissions is compared to what? All these authors cited in table 7 showed these gas reductions compared to conventional HMA. Please clarify.
[Comment #28] [Pages 13, Lines 390-391] Please elaborate on how it is easier to achieve the required densities with WMA in most cases than HMA.
[Comment #29] [Pages 13, Lines 391-394] Literature did not go through the operation and maintenance of facilities or plants used for WMA production. Therefore, please elaborate and analyze the overall cost analysis after considering the additional care needed for the WMA plants.
Author Response
Thank you for your positive, fruitful comments and suggestions, which have improved the quality of our manuscript. Please find below the revision report for your attention and perusal. The responses have been arranged based on your feedback in the review process.

Reviewer 2 Report
1)Errors in reference, such as line 787, case usage is not standard. Line 528, line 780 and line 761, appear two years.
2) The use of italics in references is confusing.
Author Response

(The authors gave the same response as above.)

Round 2
Reviewer 1 Report
· The section order is not correct, please fix it. The paper should highlight the challenges, motivations, and research gaps in using both HMA and WMA technologies in the introduction part, not by the end of the paper.
· Please fix the error of referencing the figures and tables in the whole manuscript.
Author Response
Thank you for your positive, fruitful comments and suggestions, which have improved our manuscript's quality. Please find below is the revision report for your attention and perusal. The responses have been arranged based on your feedback in the review process.
- The section order is not correct, please fix it.
Response: The sections order was checked and revised as suggested
- The paper should highlight the challenges, motivations, and research gaps in using both HMA and WMA technologies in the introduction part, not by the end of the paper.
Response: Revised as reviewer suggested. Please refer to the revised manuscript (Introduction,
Lines 92 – 118)
- Please fix the error of referencing the figures and tables in the whole manuscript
Response: Revised as suggested.
